# The Pleiotropic Effects of YBX1 on HTLV-1 Transcription

**DOI:** 10.3390/ijms241713119

**Published:** 2023-08-23

**Authors:** Susan Smith, Jaideep Seth, Amanda Midkiff, Rachel Stahl, Yu-Ci Syu, Nikoloz Shkriabai, Mamuka Kvaratskhelia, Karin Musier-Forsyth, Pooja Jain, Patrick L. Green, Amanda R. Panfil

**Affiliations:** 1Center for Retrovirus Research, Department of Veterinary Biosciences, College of Veterinary Medicine, The Ohio State University, Columbus, OH 43210, USA; smith.13211@osu.edu (S.S.); seth.70@osu.edu (J.S.); midkiff.43@osu.edu (A.M.);; 2Center for Retrovirus Research, Department of Chemistry and Biochemistry, Center for RNA Biology, The Ohio State University, Columbus, OH 43210, USA; syu.7@osu.edu (Y.-C.S.); musier@chemistry.ohio-state.edu (K.M.-F.); 3Division of Infectious Diseases, Anschutz Medical Campus, University of Colorado School of Medicine, Aurora, CO 80045, USA; nikoloz.shkriabai@cuanschutz.edu (N.S.); mamuka.kvaratskhelia@cuanschutz.edu (M.K.); 4Department of Microbiology and Immunology, Drexel University College of Medicine, Philadelphia, PA 19129, USA; pj27@drexel.edu; 5Center for Retrovirus Research, Comprehensive Cancer Center and Solove Research Institute, Department of Veterinary Biosciences, College of Veterinary Medicine, The Ohio State University, Columbus, OH 43210, USA; green.466@osu.edu

**Keywords:** HBZ, HTLV-1, latency, Tax, transcription, YBX1

## Abstract

HTLV-1 is an oncogenic human retrovirus and the etiologic agent of the highly aggressive ATL malignancy. Two viral genes, *Tax* and *Hbz*, are individually linked to oncogenic transformation and play an important role in the pathogenic process. Consequently, regulation of HTLV-1 gene expression is a central feature in the viral lifecycle and directly contributes to its pathogenic potential. Herein, we identified the cellular transcription factor YBX1 as a binding partner for HBZ. We found YBX1 activated transcription and enhanced Tax-mediated transcription from the viral 5′ LTR promoter. Interestingly, YBX1 also interacted with Tax. shRNA-mediated loss of YBX1 decreased transcript and protein abundance of both Tax and HBZ in HTLV-1-transformed T-cell lines, as well as Tax association with the 5′ LTR. Conversely, YBX1 transcriptional activation of the 5′ LTR promoter was increased in the absence of HBZ. YBX1 was found to be associated with both the 5′ and 3′ LTRs in HTLV-1-transformed and ATL-derived T-cell lines. Together, these data suggest that YBX1 positively influences transcription from both the 5′ and 3′ promoter elements. YBX1 is able to interact with Tax and help recruit Tax to the 5′ LTR. However, through interactions with HBZ, YBX1 transcriptional activation of the 5′ LTR is repressed.

## 1. Introduction

Human T-cell leukemia virus type 1 (HTLV-1) is an oncogenic retrovirus that infects an estimated 5–10 million people globally [1,2]. Unlike human immunodeficiency virus type 1 (HIV-1), HTLV-1 maintains a persistent viral infection through clonal expansion of infected T-cell clones rather than through de novo infection. Approximately 10% of infected individuals develop disease after a clinical latency period of up to several decades [1]. Diseases resulting from HTLV-1 infection include an aggressive CD4+ T-cell malignancy called adult T-cell leukemia/lymphoma (ATL) [3,4,5] and a debilitating, progressive neurologic disorder called HTLV-1-associated myelopathy/tropical spastic paraparesis (HAM/TSP) [6,7]. The precise mechanism of HTLV-1 persistence and disease development remains poorly defined. Further exploration of virus-host interactions that facilitate viral persistence and the emergence of malignant cell clones and disease is critical.

Two viral genes that are essential for the HTLV-1 lifecycle and persistence are *Tax* (transactivator from the X region) and *Hbz* (HTLV-1 basic leucine zipper factor). Tax is required for cellular transformation and drives transcription from the long terminal repeat (LTR) promoter and thus all viral genes encoded on the sense strand of the genome [8]. Tax does not directly bind DNA but instead recruits transcription factors such as CRE-binding protein (CREB)/activating transcription factor (ATF) to specific regions of the 5′ LTR [9,10]. Together, this nucleoprotein complex can recruit additional transcriptional co-activators and histone acetylases such as CREB-binding protein (CBP) and p300 to the 5′ LTR to potently activate transcription [11,12]. Conversely, the anti-sense-derived protein HBZ counteracts Tax transcriptional effects and promotes proliferation of infected cells [13,14,15,16]. Frequently, HBZ antagonism of Tax transcriptional effects is a consequence of competitive binding to cellular proteins, such as CBP/p300 [17,18]. HBZ is also essential for viral infectivity and persistence in immunocompetent rabbits, and its expression plays a critical role in infected cell survival and tumorigenesis in vivo [14,19,20].

In the pathogenesis of HTLV-1-induced ATL, there is a complete loss of 5′ LTR transcriptional activity due to mutation, deletion, and/or hypermethylation of the DNA sequences [21,22]. Despite the frequent loss of 5′ LTR transcriptional activity, the 3′ LTR (which encodes for HBZ) remains transcriptionally active in cell lines, patients infected with HTLV-1, and ATL disease patients [13]. Transcription from the 3′ LTR antisense promoter is regulated by factor SP1 (specificity protein 1), JunD, and MEF-2C (myocyte enhancer factor 2C) binding sites located in the promoter region [13,23,24,25,26,27]. Recent reports have suggested the 3′ region of the provirus remains transcriptionally active in part due to a viral CTCF (CCCTC-binding factor) binding site [28] and an internal enhancer element [29,30].

While *hbz* is the only viral transcript constitutively expressed, *tax* has been shown to have transient, spontaneous expression in ATL cells [22,31,32,33]. This “on/off switch” in Tax expression only occurs in a portion of infected cells at any given time but is crucial for maintaining the entire cell population. Hence, the regulation of sense and antisense transcription is thought to have critical importance in long-term HTLV-1 infection, latency, and leukemogenesis [34]. However, the mechanism of this transcriptional regulation remains elusive, and we hypothesize that host protein interactions may play a crucial role. 

Herein, we have identified the Y-box binding protein (YBX1, YB-1, nuclease-sensitive element-binding protein 1, or NSEP1) as a novel HBZ-interacting protein. YBX1 is a DNA- and RNA-binding protein with diverse functions, including modulation of transcription and translation, DNA repair, and pre-mRNA splicing [35,36]. YBX1 has also been recognized as an oncoprotein and is overexpressed in many different cancer types, including lymphoproliferative neoplasms [37,38]. In addition to driving the proliferation of normal T cells, YBX1 can regulate the lifecycle of several different viruses, including HIV-1, dengue virus, influenza, herpes simplex virus type 1, murine leukemia virus, and hepatitis C virus [39,40,41,42,43,44,45]. In retroviruses such as HIV-1, YBX1 is a cofactor in multiple stages of the viral lifecycle. It not only binds the viral promoter and increases transcriptional activity via interactions with the HIV-1 Tat-TAR transcriptional complex, but it also interacts with integrase and supports both early and late viral replication [35,46]. YBX1 can also stabilize newly synthesized viral genomic RNA to enhance the production of HIV-1 [47]. Limited studies have investigated the role of YBX1 in the HTLV-1 lifecycle. In 1994, Kashanchi et al. identified a cellular protein that bound a small portion (45 nt) of the HTLV-1 LTR [48]. Based on amino acid sequencing, this protein was most homologous to the C-terminal amino acids of YBX1. In this study, co-transfection of YBX1 and HTLV-1 LTR reporter plasmids suggested that YBX1 may regulate transcription. However, the impact that YBX1 has on the 3′ LTR and the role of HBZ as a YBX1 binding partner have yet to be established. 

Given our current understanding of YBX1 and HBZ, we hypothesized that the HBZ/YBX1 interaction plays a key role in viral gene expression and HTLV-1 pathobiology. We found that YBX1 activates transcription from the 5′ LTR promoter, interacts with Tax, and together with Tax, can synergistically activate 5′ LTR transcription. Conversely, Hbz interacts with YBX1 and can inhibit YBX1-mediated transcriptional activation of the 5′ LTR. Using chromatin immunoprecipitation (ChIP) assays, we found YBX1 associates with both the 5′ and 3′ LTRs, driving both sense and antisense gene expression, respectively. Finally, loss of YBX1 using shRNA-mediated knockdown decreased both Tax and Hbz expression in HTLV-1-transformed and ATL-derived T-cell lines. Together, this data suggests YBX1 positively influences transcription from both the 5′ and 3′ LTR promoter elements; however, through interactions with HBZ, YBX1 transcriptional activation of the 5′ LTR is repressed. In summation, YBX1 exerts pleotropic effects on viral transcription and regulates viral protein expression, thus contributing to the control of viral latency.

## 2. Results

### 2.1. HBZ Interacts with the Cellular Factor YBX1

To determine if cellular binding partners of HBZ are critical to pathogenesis, we identified cellular proteins that preferentially bind to HBZ but not the HTLV-2 homolog APH-2 (the antisense protein of HTLV-2). HTLV-2 is a closely related, nonpathogenic retrovirus not strongly linked to disease [49]. APH-2 has less than 30% homology to HBZ; however, these antisense-derived proteins do share some functional similarities [50]. Using affinity capture coupled with shotgun proteomics, the YBX1 protein was identified as a potential interacting partner of HBZ (Appendix A). In addition to YBX1, several previously described HBZ-interacting factors, including AP1, ILF3, JunD, XRCC6, JunB, and XRCC5, were also identified (Figure 1A). Affinity capture pulldown assays using S-tagged HBZ and APH-2 in HEK293T cells confirmed that YBX1 interacts with HBZ (Figure 1B). Using more physiologically relevant HTLV-1-infected T-cell lines, protein-protein interaction between YBX1 and HBZ was examined by co-immunoprecipitation (IP). In the newly immortalized T-cell line, PBL-HTLV-1, and the HTLV-1-transformed T-cell line, SLB-1, we identified YBX1 protein when immunoprecipitating with an HBZ antibody, but not immunoglobulin control (Figure 1C). PBL-HTLV-1 and SLB-1 cells express both sense and antisense viral proteins, while ATL-derived T-cell lines ATL-ED and TL-Om1 only express HBZ. We also identified the YBX1 protein when immunoprecipitating HBZ in ATL-derived cell lines (Figure 1D).

To identify regions of the HBZ or YBX1 protein responsible for interaction, a series of HBZ and YBX1 deletion mutants (Appendix A, respectively) were used together with affinity capture pulldown assays. HBZ is comprised of an N-terminal transactivation domain (AD), a central basic region (BR2/1/3), and a C-terminal leucine zipper domain (bZIP). YBX1 was able to interact with the HBZ AD-BR2/1/3 and BR/bZIP mutants but not the HBZ bZIP and AD-BR2/1 mutants (Appendix A). This suggests the BR3 region of HBZ is critical for interaction with YBX1. The central domain of HBZ is comprised of three motifs (BR2/1/3), which together are responsible for the nuclear localization of HBZ [51]. The YBX1 protein is comprised of a variable N-terminal alanine/proline-rich domain (A/P), a central cold shock domain (CSD), and a C-terminal domain (CTD). HBZ was able to interact with the YBX1 ∆A/P, ∆CSD, and CTD domains but not the ∆CTD domain (Appendix A). This suggests the CTD of YBX1 is critical for interaction with HBZ. YBX1 CTD is intrinsically disordered and has been implicated in various nucleic acid binding and protein-protein interactions [36,52]. 

### 2.2. YBX1 Activates Viral Transcription

Previous studies identified YBX1 as a binding factor that may influence basal 5′ LTR transcription [48]. Given the interaction between YBX1 and HBZ, we asked if YBX1 could activate transcription and what effect HBZ would have on YBX1 transcriptional activation. Using dual luciferase reporter assays, HEK293T cells were transfected with increasing amounts of the plasmid encoding YBX1, and transcription from the viral 5′ LTR promoter was measured (Figure 2A). In the presence of HBZ, we found LTR transcriptional activation was significantly decreased at the lowest transfected amount of YBX1, suggesting HBZ can inhibit YBX1-mediated LTR activation. Conversely, when YBX1 protein levels were decreased using shRNA-mediated knockdown, we found a significant decrease in basal 5′ LTR transcriptional activation (Figure 2D). Since Tax is the main transcriptional activator of the 5′ LTR, we next examined what effect Tax and YBX1 together would have on viral transcription. Using reporter gene assays, we found a synergistic positive effect on transcription when cells were transfected with plasmids encoding Tax and increasing amounts of YBX1 (Figure 2B). We also found Tax-mediated transcriptional activation was repressed when YBX1 protein levels were decreased in HEK293T cells, further suggesting Tax transcriptional activation may be mediated in part through YBX1 (Figure 2E). Tax transcriptional activation was also significantly repressed in shYBX1 Jurkat T cells (Figure 2F). HBZ can repress Tax-mediated transcription [15,16,17], therefore we next asked if HBZ could also repress the Tax/YBX1 synergistic transcriptional activation (Figure 2C). Using reporter gene assays, we observed a significant decrease in Tax/YBX1 transcriptional activation in the presence of HBZ, suggesting the HBZ/YBX1 interaction may interfere with Tax/YBX1 transcriptional activation. 

### 2.3. YBX1 Interacts with Tax 

The synergistic transcriptional effect of YBX1 and Tax prompted us to ask whether YBX1 could interact with Tax (in addition to HBZ). Using PBL-HTLV-1 and SLB-1 T-cell lines, we found YBX1 was enriched when either Tax or HBZ was immunoprecipitated but not when using normal rabbit IgG (Figure 3A,B). When Tax was immunoprecipitated, we also observed a small amount of HBZ protein that was pulled down. This may be due to a multi-protein complex formed between HBZ, YBX1, and Tax. To determine if Tax can interact with YBX1 in the absence of HBZ, we utilized a Tax-inducible Jurkat T-cell line called JPX-M. JPX-M cells contain a metallothionein promoter-driving Tax expression plasmid [53]. Due to the well-known cytotoxicity of over-expressed Tax protein, we chose JPX-M cells, which have a simple in-frame insertion of a single arginine residue at position 63 that renders Tax non-functional and less toxic to T-cell growth and survival [54]. YBX1 immunoprecipitation, after cadmium chloride induction of Tax, confirmed that Tax can bind YBX1 in the absence of HBZ (Figure 3C). HBZ antagonism of Tax is frequently a consequence of competitive binding to cellular proteins, such as CBP/p300 [17,18]. Therefore, the ability of YBX1 to bind both Tax and HBZ suggests competitive antagonism of LTR transcription.

### 2.4. YBX1 Activates Sense and Antisense Viral Transcription 

Given the similarity between the 5′ and 3′ LTR sequences, we asked if YBX1 could also activate antisense transcription from the 3′ LTR. HEK293T cells were co-transfected with an HTLV-1 proviral clone in the presence and absence of the YBX1 expression vector. We found an increase in HBZ protein expression (Figure 4A) and *hbz* transcript (Figure 4B) in the presence of exogenous YBX1. Conversely, when endogenous YBX1 was decreased using shRNA, we found a decrease in Tax protein (Figure 4C), a decrease in both *tax* and *hbz* transcripts (Figure 4D), and a decrease in p19 Gag release into the supernatant (Figure 4E). Finally, YBX1 knockdown in HTLV-1-transformed SLB-1 T cells and ATL patient-derived ATL-ED T cells resulted in a decrease in HBZ protein and Tax protein (only expressed in SLB-1) (Figure 4F,G). Tax protein was not detected in ATL-ED cells since these cells have a hypermethylated 5′ LTR and do not express measurable Tax protein. Taken together, these results suggest YBX1 activates viral transcription from both the 5′ and 3′ LTR promoters. 

### 2.5. YBX1 Associates with the 5′ and 3′ Viral LTR 

We performed chromatin immunoprecipitation (ChIP) assays in HTLV-1-positive T cells to determine whether YBX1 can bind the 5′ and/or 3′ LTR of HTLV-1. We chose two HTLV-1 cell lines that have active transcription from both the 5′ and 3′ LTRs and express all the viral genes (PBL-HTLV-1 and SLB-1) and two ATL-derived cell lines that have a hypermethylated 5′ LTR and subsequent silent sense transcription (TL-Om1 and ATL-ED). We found that YBX1 was associated with both the 5′ LTR (Figure 5A) and the 3′ LTR (Figure 5B) in all HTLV-1 cell lines but not with the GAPDH gene. Interestingly, we observed less YBX1 binding to the 5′ LTR in ATL-derived cell lines. We attribute this to the hypermethylated nature of the 5′ LTR or, alternatively, the number of integrated proviral genomes in these cells.

Previous studies using gel-shift assays suggested there may be a YBX1 binding site within a short region of the viral 5′ LTR from +195 to +240 [48]. This study identified an imperfect YBX1 binding site (Figure 6A) in the viral 5′ LTR. To determine if YBX1 directly contacts the DNA at this specific location, we generated a YBX1 binding site mutant in the LTR-luciferase reporter vector (YBX1m LTR). To evaluate whether mutation of the hypothesized YBX1 binding site in the LTR abolishes YBX1 transcriptional activation, HEK293T cells were transfected with wild-type (wt) or YBX1m LTR-1-luciferase constructs with or without a YBX1 expression vector (Figure 6B). YBX1 was able to activate transcription of both the wt and YBX1m LTR reporter constructs at equivalent levels. In addition, YBX1m had no effect on the ability of Tax to activate transcription or HBZ to repress transcription (Figure 6C). Finally, decreased YBX1 expression (via shRNA) repressed Tax transcriptional activation with both the wt and YBX1m LTR luciferase constructs (Figure 6D). This data suggests that although YBX1 can associate with the viral LTR, it may not bind DNA directly or there may be more than one YBX1 binding site present in the LTR.

### 2.6. YBX1 Enhances Tax Binding and HBZ Represses YBX1 Transcriptional Activation

To further understand what effect YBX1 has on Tax association with the 5′ LTR, we examined Tax binding in the presence and absence of YBX1 using a ChIP assay. Using the HTLV-1-transformed SLB-1 T-cell line, we found Tax association with the 5′ LTR was significantly decreased when YBX1 protein levels were reduced (Figure 7A). Conversely, to determine what effect HBZ has on YBX1 LTR transcriptional activation, we next examined the ability of YBX1 to activate transcription in the presence and absence of HBZ using our wt.HTLV-1 proviral plasmid and a molecular clone of HTLV-1 that lacks HBZ protein expression (HTLV-1.∆HBZ). We found that YBX1 significantly increased LTR activation (Figure 7B) and increased *hbz* transcript levels (Figure 7C) in the absence of HBZ protein. This data suggests HBZ represses YBX1 transcriptional activation. Conversely, YBX1 protein knockdown (via shRNA) decreased steady-state Tax protein expression in the presence of HBZ (Figure 7D). However, YBX1 protein knockdown did not affect Tax protein expression in the absence of HBZ. This data implies that HBZ helps repress Tax transcriptional activation of the 5′ LTR (and thus Tax protein expression) through YBX1. 

## 3. Discussion

HTLV-1 gene expression plays a key role in viral persistence and pathogenesis. The integrated viral genome (provirus) is flanked by LTRs at both the 5′ and 3′ ends. The LTRs contain promoters responsible for both sense (5′ LTR) and antisense (3′ LTR) viral transcription. Several studies have shown that viral genes, such as *Tax* and *Hbz*, play an important role in the viral lifecycle [55]. Tax is a multi-functional, highly immunogenic protein that is critical during the early stages of HTLV-1 infection as it promotes transactivation of the 5′ LTR to drive the transcription of all plus-strand viral transcripts [56,57,58,59]. Tax can also activate cell signaling pathways that are critical for viral transformation, most notably the NF-κB pathway [60,61]. The oncogenic potential of Tax is partially derived from its ability to repress DNA damage repair pathways, subsequently causing deregulation of the cell cycle and a higher propensity for detrimental mutations in HTLV-1-infected cells [62]. These infected T cells may also undergo cellular senescence or apoptosis induction due to Tax-mediated hyperactivation of NF-κB [63,64,65,66], necessitating regulation of Tax expression throughout the course of infection [67]. Tax expression is typically low or undetectable in most ATL cells. However, recent evidence has revealed that Tax is expressed in a minor fraction of leukemic cells at any given time, and this expression is spontaneously switched between “on” and “off” states [68]. This transient Tax expression is critical for maintaining the infected cell population. 

Hbz is transcribed from the 3′ LTR and also counteracts many functions of Tax including 5′ LTR transactivation and NF-κB activation. Transient silencing of Tax expression by Hbz is vital for the virus to successfully regulate viral transcription, evade immune detection, and maintain viral latency [15,17]. Interestingly, *Hbz* is the only viral gene that remains intact and is consistently expressed in all ATL cases [13]. Previous work has shown that shRNA-mediated Hbz knockdown in leukemic cells correlates with a significant decrease in T-cell proliferation in culture [19]. Engraftment of these leukemic cells in NOD.Cg-PrkdcSCIDIL2rgtm1Wjl/SzJ (NOG) mice will form solid tumors that also infiltrate multiple tissues. However, leukemic cells knocked down for Hbz showed significantly decreased tumor formation and organ infiltration compared to animals inoculated with wild-type cells. Infection of rabbits with the HBZ-deficient virus significantly reduces proviral load and viral persistence, while infection of humanized mice with the same virus significantly decreases lymphoproliferative disease development [14,20]. These data confirm Hbz expression enhances the proliferative capacity of HTLV-1-infected T cells and plays a critical role in cell survival and tumorigenesis. However, recent data has also suggested Hbz protein may promote apoptosis, while *hbz* RNA induces proliferation [69]. Tight regulation of HTLV-1 gene expression is critical throughout the course of infection. The combined action of Tax and HBZ enables the virus to immortalize infected target cells, avoid immune detection, and establish persistent infection. A better understanding of the factors that regulate Tax and Hbz expression will provide additional therapeutic targets for the treatment of HTLV-1 and associated diseases. 

Using affinity capture coupled with shotgun proteomics, we have identified numerous cellular factors that interact with the HBZ protein, including the cellular transcription factor YBX1. We further confirmed our proteomics analysis using affinity pulldowns in HEK293T cells with HBZ over-expression vectors and in HTLV-1-transformed T cells. While we were able to immunoprecipitate HBZ and demonstrate YBX1 binding in these assays, we were unsuccessful in reciprocal immunoprecipitations. This could be due to the molecular weight of HBZ being nearly identical to that of the immunoglobulin light chain (thus making protein detection difficult) or because the antibody used to immunoprecipitate YBX1 interfered with HBZ binding. HBZ deletion mutants identified the HBZ/YBX1 region of interaction within the central BR3 region, which is important for nuclear localization of HBZ. The region of YBX1 responsible for interaction with HBZ was identified as the CTD domain—a large domain responsible for many other protein-protein interactions, including IRP2, hnRNP K, and YBAP1 [70,71,72]. To examine the importance of the YBX1/HBZ interaction on pathogenesis, future experiments utilizing single-point mutations that abolish this interaction or small-molecule inhibitors that disrupt this binding would be highly beneficial. 

A previous study analyzed a 45-nucleotide (nt) sequence from the 5′ LTR that was important for basal transcription [48]. Using a lambda gt11 Jurkat expression library, a cellular factor with homology to YBX1 that bound the 45-nt LTR sequence in vitro was identified. Using gel shift assays, a potential YBX1 binding site was also identified within the LTR. Although YBX1 was previously implicated in HTLV-1 transcription, we thought it prudent to reanalyze the effect of YBX1 in the context of HBZ given the observed protein-protein interaction. We confirmed that YBX1 can activate transcription from the 5′ LTR. We also found that YBX1 and Tax can synergistically activate transcription, while HBZ serves to repress this synergy. Importantly, our results were found using both over-expression vectors and shRNA-mediated protein knockdown in both HEK293T and the HTLV-1 physiologically relevant Jurkat T-cell line. Interestingly, the near complete loss of Tax transcriptional activation in the absence of YBX1 in Jurkat cells suggests the importance of YBX1 transcriptional activation in a T-cell environment. 

HBZ protein is able to suppress Tax-mediated transcription through competitive binding and subsequent blocking of LTR recruitment of cellular factors such as ATF/CREB and CBP/p300. We found that, like HBZ, Tax is also able to interact with YBX1 in both HTLV-1-transformed T cells (which express other viral proteins) and in Jurkat T cells in the absence of other viral proteins. shRNA-mediated knockdown of YBX1 in both HEK293T and naturally infected T-cell lines resulted in decreased Tax and HBZ protein and transcript, suggesting YBX1 can activate transcription from the 5′ as well as antisense transcription from the 3′ LTR. We found YBX1 can associate with both the 5′ and 3′ LTRs in HTLV-1-transformed cells, although mutation of the previously identified YBX1 binding site [48] had no effect on LTR transcription. This suggests that YBX1 may not bind the DNA directly but may associate with it through other cellular factors. Alternatively, more than one YBX1 binding site may be present in the viral LTR. Finally, we were able to demonstrate that loss of YBX1 decreases Tax association with the 5′ LTR, and the absence of HBZ increases YBX1-mediated activation of the 5′ LTR. Taken together, our results suggest a model whereby YBX1 associates with the 5′ and 3′ viral LTRs to activate transcription (Figure 8). YBX1 recruits Tax to the 5′ LTR to synergistically activate transcription; HBZ (in addition to squelching CBP/p300) competitively binds to YBX1, thus removing it from the viral LTR and decreasing transcription. 

YBX1 can bind DNA and RNA and has many diverse roles and functions within the cell, ranging from transcription and translation to DNA repair. Indeed, recent studies have shown that YBX1 may have a critical role in leukemogenesis in non-infectious T-cell acute lymphoblastic leukemia (T-ALL) by regulating expression of the PI3K/AKT and ERK signaling pathways [73]. In addition, in vitro and in vivo data using YBX1 inhibitors suggests that these drugs may be an effective treatment for T-ALL [73]. We have analyzed only one facet of YBX1 function (transcription) within HTLV-1 infection. Future studies should examine YBX1 effects on other aspects of the viral lifecycle (splicing, assembly) and determine if YBX1 interaction can affect other known functions of Tax and HBZ. Conversely, the effect of HBZ and/or Tax on normal YBX1 functions may provide new insights into HTLV-1 pathogenesis. Further characterization of factors that regulate (promote or repress) viral transcription will provide insight into HTLV-1 gene expression, persistence, and oncogenesis. 

## 4. Materials and Methods

### 4.1. Cell Culture

HEK293T cells were maintained in Dulbecco’s modified Eagle’s medium (DMEM) supplemented with 10% fetal bovine serum (FBS; Gemini Bio-Products, West Sacramento, CA, USA), 2 mM glutamine, penicillin (100 U/mL), and streptomycin (100 μg/mL). ATL-ED, TL-Om1, Jurkat, and JPX-M cells were maintained in RPMI 1640 medium supplemented with 10% FBS, 2 mM glutamine, 100 U/mL penicillin, and 100 μg/mL streptomycin. JPX-M cells are a subclone of Jurkat cells with a metallothionein promoter driving a Tax point mutation (transcriptionally inactive) expression plasmid [53]. PBL-HTLV-1 cells were maintained in RPMI 1640 medium supplemented with 20% FBS, 2 mM glutamine, 100 U/mL penicillin, 100 μg/mL streptomycin, and 20 units of hIL-2/mL. SLB-1 cells were maintained in Iscove’s Modified Dulbecco’s medium supplemented with 10% FBS, 2 mM glutamine, penicillin (100 U/mL), and streptomycin (100 μg/mL). All cells were grown at 37 °C in a humidified atmosphere of 5% CO_2_ and air.

### 4.2. Plasmids 

Plasmid DNA was purified on maxi-prep columns according to the manufacturer’s protocol (Qiagen, Valencia, CA, USA). The S-tagged APH-2, S-tagged HBZ, S-tagged Tax, and pME-HBZ expression vectors were generated and described previously [14,50,74]. The pcDNA™3.1(+) negative control (empty vector) was purchased from Invitrogen Life Technologies (Carlsbad, CA, USA), and the S-tagged empty vector (pTriExTM-4 Neo) was purchased from Millipore Sigma (St. Louis, MO, USA). S-tagged HBZ domain mutants (AD-BR2-BR1-BR3, BR/bZIP, bZIP, and AD-BR2/1) were previously described [75]. pDestmycYBX1 was a gift from Thomas Tuschl (Addgene plasmid#19878; http://n2t.net/addgene:19878; RRID:Addgene_19878; access date 1 August 2019) [76]. The N-terminal myc-tagged YBX1 coding sequence in pDestmycYBX1 was subcloned into the pCMV3 mammalian expression plasmid (Sino Biological, Beijing, China). The final plasmid was named pCMV3 YBX1. All constructs for truncation mutants of YBX1 were generated by the NEB builder HiFi DNA assembly kit (New England Biolabs, Ipswich, MA, USA) according to the manufacturer’s protocol (Table 1). 

The wild-type HTLV-1 molecular clone (ACHneo) was previously described [77]. HTLV-1.∆Hbz contains a G to A point mutation that terminates the HBZ reading frame at amino acid 8 of the HBZ major transcript and was previously described [14]. The HTLV-1 LTR-luciferase reporter plasmid (LTR-1-Luc) and thymidine kinase (tk)-Renilla transfection efficiency control plasmid were described previously [78]. The YBX1m LTR-luciferase reporter plasmid was generated using site-directed mutagenesis, and mutations were selected based on a previously proposed YBX1 binding site in the HTLV-1 LTR [48]. 

### 4.3. S-Tag-Affinity Pulldown Assay

HEK293T cells were transfected with the indicated expression vectors using Lipofectamine^®^ 2000 (Invitrogen) according to the manufacturer’s instructions. Cells were washed 24 h post-transfection with 1× PBS, and lysates were prepared with NP-40 lysis buffer in the presence of a protease inhibitor (Millipore Sigma). Cells were centrifuged at maximum speed (16,300× *g*) for 10 min at 4 °C. S-tag purification was performed by rocking cell lysates with S beads (Millipore Sigma) overnight at 4 °C. The S-tag beads were washed twice with NP-40 lysis buffer. An equal volume of 2× SDS-sample buffer was added, and proteins were extracted by heating at 95 °C for 10 min.

### 4.4. Mass Spectrometry and Proteomic Analysis

Affinity capture (S-tag) coupled with shotgun proteomics was performed as previously described [75,79]. Briefly, S-tag pull-down samples were subjected to SDS-PAGE, stained using GelCode Blue Stain Reagent (Thermo Fisher Scientific, Waltham, MA, USA), and whole lanes were excised from the gel and subjected to proteomics analysis. The gel pieces were de-stained with 50% acetonitrile and then subjected to in-gel proteolysis using sequencing-grade modified trypsin (Promega, Madison, WI, USA). The resulting peptides were extracted in acetonitrile by vortexing for 10 min and then desiccated in an Eppendorf^®^ Vacufuge^®^ Plus Vacuum Concentrator. The samples were run on a Thermo Scientific Q Exactive mass spectrometer and analyzed with Mascot software (Matrix Science, Boston, MA, USA). The data were visualized using Scaffold Viewer version 4.8.9 (Proteome Software, Inc., Portland, OR, USA), with a probability of >95% and a minimum of 2 peptides.

### 4.5. Immunoprecipitation

Cells were washed with 1× PBS and incubated with gentle rocking at 4 °C for 30 min in NP-40 lysis buffer (150 mM NaCl, 1% NP-40, 50 mM Tris-Cl pH 8.0). Cells were centrifuged at maximum speed (16,300× *g*) for 10 min at 4 °C. Lysates were rocked overnight at 4 °C with 2 µg of the following antibodies: control rabbit IgG (Santa Cruz Biotechnology, Dallas, TX, USA), Tax [80], HBZ [14], or YBX1 (Bethyl Laboratories, Montgomery, TX, USA; A303-231A). Immune complexes were captured using Dynabeads Protein G and washed three times in NP-40 lysis buffer. 2× SDS-sample buffer was added, and proteins were extracted by heating at 95 °C for 10 min. Precipitated proteins were examined using immunoblotting. VeriBlot for IP detection (Abcam, Cambridge, MA, USA) was used as a secondary antibody to reduce background from immunoglobulin heavy and light chains.

### 4.6. Transfections, Luciferase Reporter Assays, and p19 Gag ELISA

HEK293T were transfected using Lipofectamine^®^ 2000 (Invitrogen Life Technologies) according to the manufacturer’s protocol. Jurkat cells were transfected with Lipofectamine^®^ 2000 or Lipofectamine^®^ LTX (Invitrogen Life Technologies) according to the manufacturer’s instructions. Luciferase assays were performed by lysing cell pellets in 1X Passive Lysis Buffer (Promega) and following the manufacturer’s protocol for the Luciferase Assay System (Promega). Firefly luciferase and renilla luciferase relative light units were measured using the FilterMax F5 Multi-Mode Microplate Reader (Molecular Devices, San Jose, CA, USA). Cell supernatants were collected 48 h post-transfection to measure HTLV-1 p19 Gag production using the RETRO-TEK HTLV p19 Antigen ELISA (ZeptoMetrix Corporation, Buffalo, NY, USA) and following the manufacturer’s instructions.

### 4.7. Immunoblotting

Cells were lysed in NP-40 lysis buffer containing a protease inhibitor cocktail (Millipore Sigma) and quantitated using a Pierce bicinchoninic acid protein assay kit (Thermo Fisher Scientific). Equivalent amounts of protein were separated in Mini-Protean TGX precast 4 to 20% gels (Bio-Rad Laboratories, Hercules, CA, USA) and transferred to nitrocellulose membranes. Membranes were blocked in phosphate-buffered saline (PBS) containing 5% milk plus 0.1% Tween^®^ 20 and incubated with the primary antibody. The following primary antibodies were used: HBZ (1:1000) [14], APH-2 (1:1000) [74], S-tag (1:1000; Abcam), Tax (1:1000), YBX1 (1:5000, Bethyl A303-231A), myc (1:1000; Abcam ab32), and β-actin (1:5000; Millipore Sigma). The secondary antibodies used were horseradish peroxidase-labeled goat anti-rabbit and goat anti-mouse immunoglobulin antibodies (1:5000; Santa Cruz Biotechnology). Membranes were developed using Pierce ECL Western Blotting Substrate (Thermo Fisher Scientific) and imaged using an Amersham Imager 600 (GE Healthcare Life Sciences, Piscataway, NJ, USA). Densitometric data were calculated using the ImageQuant TL program (GE Healthcare Life Sciences).

### 4.8. Lentiviral Production and Cell Transduction

Lentiviral vectors expressing human YBX1-directed short hairpin RNAs (shRNA) (Dharmacon Reagents, Lafayette, CO, USA) and a negative control pLKO.1 (rH24080) were propagated according to the manufacturer’s instructions. To generate lentivirus, HEK293T cells were transfected with lentiviral vector(s) as well as DNA vectors encoding HIV Gag/Pol and vesicular stomatitis virus G in 10-cm dishes using Lipofectamine^®^ 200 reagent according to the manufacturer’s instructions. Media containing the lentiviral particles was collected 72 h later. Lentiviral particles were concentrated using ultracentrifugation in a Sorvall SW-41 swinging bucket rotor at 90,000× *g* for 1.5 h at 4 °C. Target cells were transduced using polybrene (8 µg/mL) and spin-inoculation at 2000× *g* for 2 h at room temperature. Three days post-transduction, cells were selected with puromycin (1 µg/mL) for 5–10 days.

### 4.9. Chromatin Immunoprecipitation (ChIP)

ChIP experiments were carried out as previously described [81]. Briefly, cells were cross-linked in fresh 1% paraformaldehyde for 10 min at room temperature. The cross-linking reaction was quenched with 125 mM glycine. Following cell lysis and DNA fragmentation by sonication, DNA-protein complexes were immunoprecipitated with YBX1, Tax, and control IgG antibodies. Immunoprecipitated DNA-protein complexes were washed using sequential low-salt, high-salt, lithium chloride, and Tris-EDTA (TE) buffers. DNA was purified using the Qiagen Gel Extraction Kit (Qiagen). The presence of specific DNA fragments in each precipitate was detected using qPCR. Primers used for amplifying the HTLV-1 5′ LTR and 3′ LTR were previously described [28].

### 4.10. Quantitative RT-PCR (qRT-PCR)

Total RNA was isolated using an RNeasy Kit (Qiagen) according to the manufacturer’s instructions. Isolated RNA was quantitated, and reverse transcription (RT) was performed using a SuperScript first-strand synthesis system for RT-PCR (Invitrogen Life Technologies) according to the manufacturer’s instructions. The final reaction volume was 20 μL and consisted of 10 μL of iQ SYBR green Supermix (Bio-Rad), 300 nM of each specific primer, and 2 μL of cDNA template. For each run, cDNA and a no-template control were all assayed in duplicate. The reaction conditions were 95 °C for 5 min, followed by 40 cycles of 94 °C for 30 s, 56 °C for 30 s, and 72 °C for 45 s. Primer pairs for the specific detection of viral mRNA species (*tax*/*rex*, *hbz*) and human glyceraldehyde-3-phosphate dehydrogenase (*hgapdh*) were described previously [75,82]. Data from triplicate experiments are presented in histogram form as means with standard deviations.

### 4.11. Statistics

Statistical analyses were performed using GraphPad Prism 7 software (GraphPad Software) as indicated. Studies were analyzed by the Student’s *t*-test. Statistical significance was defined as * *p* ≤ 0.05, ** *p* ≤ 0.01, *** *p* ≤ 0.001, and **** *p* ≤ 0.0001.

## Figures and Tables

**Figure 1 ijms-24-13119-f001:**
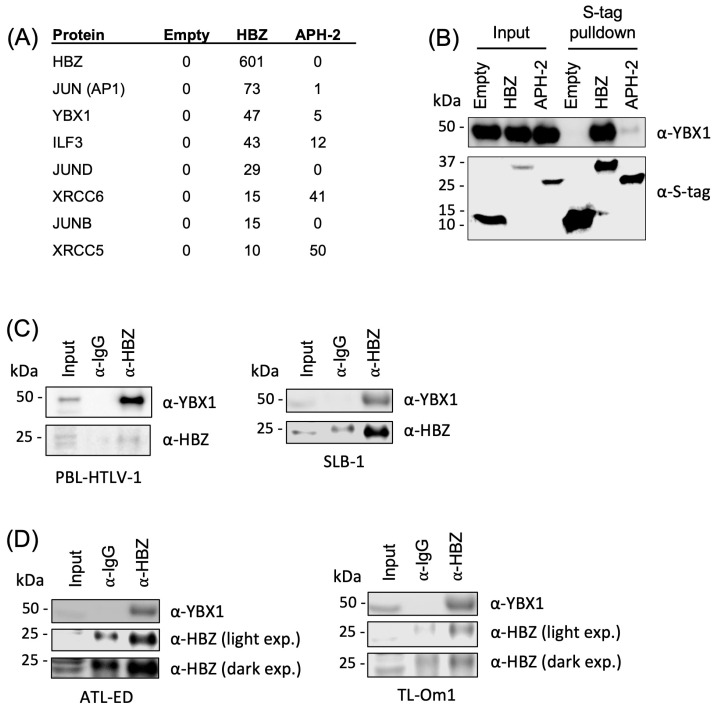
YBX1 interacts with HBZ. (**A**) MS-MS experiments were performed to identify unique protein interactions. Numbers indicate the total spectral counts, or numbers of unique peptides, found in a subset of proteins with HBZ and APH-2 pulldown. (**B**) HEK293T cells were transfected with empty, HBZ, or APH-2 S-tagged expression vectors. Proteins were purified by S-tag affinity 24 h after transfection. Pulldowns were examined by immunoblot analysis using antibodies to YBX1 and S-tag. (**C**) PBL-HTLV-1 (HTLV-1 newly immortalized), SLB-1 (HTLV-1 transformed), and (**D**) ATL-ED or TL-Om1 (ATL-derived) T-cell lines were immunoprecipitated with a control rabbit antibody or HBZ rabbit antisera. Immunoprecipitated proteins were examined by immunoblot analysis using antibodies to YBX1 and HBZ. 10% of IP was included as a direct load.

**Figure 2 ijms-24-13119-f002:**
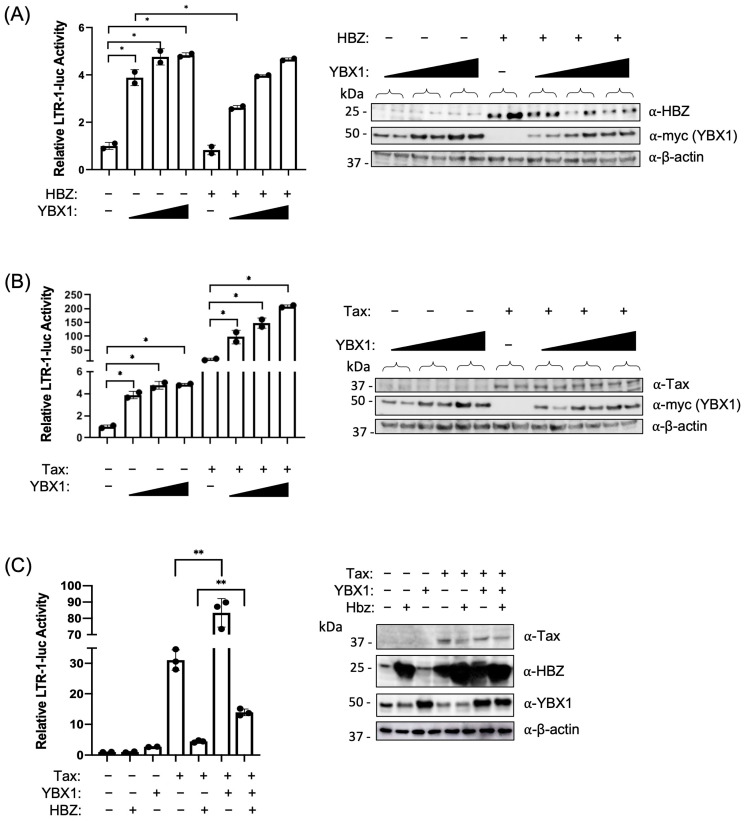
YBX1 influences viral transcription. (**A**–**C**) HEK293T cells were co-transfected with (**A**) pcDNA3.1(+) empty, HBZ, and increasing amounts of myc-tagged YBX1 expression vectors; (**B**) pcDNA3.1(+) empty, Tax, and increasing amounts of myc-tagged YBX1 expression vectors; or (**C**) pcDNA3.1(+) empty, Tax, HBZ, and/or myc-tagged YBX1 expression vectors with an HTLV-1 LTR-firefly luciferase construct and tk-renilla luciferase construct (internal control). At 48 h post-transfection, cells were collected for luciferase assays to measure relative LTR transactivation (left panels). The empty vector was set at 1. Total cell lysates were examined by immunoblot analysis using antibodies to YBX1, myc (YBX1), HBZ, Tax, and β-actin (right panels). HEK293T cells (**D**,**E**) were transduced with lentiviral vectors expressing shRNA directed against YBX1. After a brief puromycin selection, the cells were co-transfected with (**D**) an HTLV-1 LTR-firefly luciferase construct, a tk-renilla luciferase construct (internal control), and (**E**) pcDNA3.1(+) empty, Tax, and/or HBZ expression vectors. At 48 h post-transfection, cells were collected for luciferase assays to measure relative LTR transactivation (left panels). Empty vector was set at 1. Total cell lysates were examined by immunoblot analysis using antibodies to YBX1, HBZ, Tax, and β-actin (right panels). (**F**) Jurkat cells were transduced with lentiviral vectors expressing shRNA directed against YBX1. After a brief puromycin selection, the cells were co-transfected with an HTLV-1 LTR-firefly luciferase construct, a tk-renilla luciferase construct (internal control), pcDNA3.1(+) empty, Tax, and/or HBZ expression vectors. 48 h post-transfection, cells were collected for a luciferase assay to measure relative LTR transactivation (left panel). The empty vector was set at 1. Total cell lysates were examined by immunoblot analysis using antibodies to YBX1 and β-actin (right panel). Graphs represent data generated from duplicate samples, and error bars represent the standard deviation (SD). The data are representative of at least three experimental repeats. Statistical significance was determined using a Student’s *t*-test; * *p* ≤ 0.05, ** *p* ≤ 0.01, **** *p* ≤ 0.0001.

**Figure 3 ijms-24-13119-f003:**
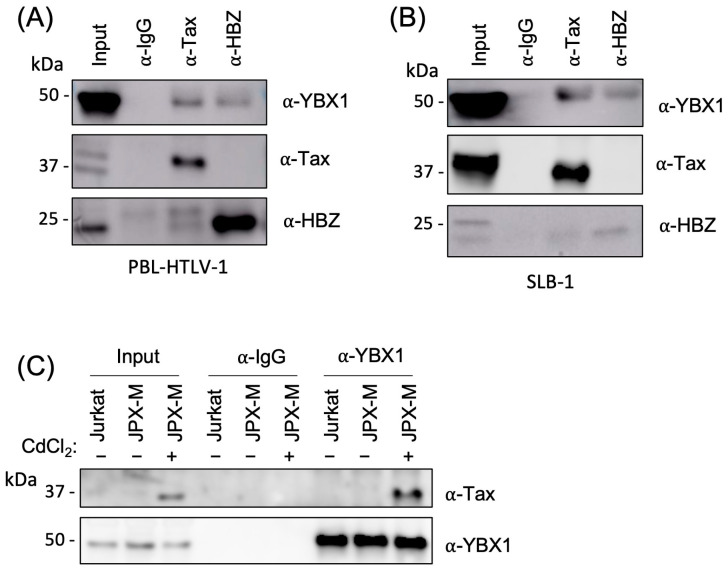
YBX1 interacts with Tax. (**A**) PBL-HTLV-1 and (**B**) SLB-1 cell lines were immunoprecipitated with a control rabbit IgG, Tax, or HBZ antibody. IPs were examined by immunoblot analysis using antibodies to YBX1, Tax, and Hbz. 10% of IP was included as a direct load. (**C**) JPX-M cells (subclone of Jurkat cells with a metallothionein promoter-driven Tax mutant expression plasmid) were treated with 15 µM cadmium chloride (CdCl_2_) for 24 h to produce Tax protein. Uninduced and induced JPX-M cells and Jurkat cells were immunoprecipitated with a control rabbit IgG or YBX1 antibody. IPs were examined by immunoblot analysis using antibodies to YBX1 and Tax. 10% of IP was included as a direct load.

**Figure 4 ijms-24-13119-f004:**
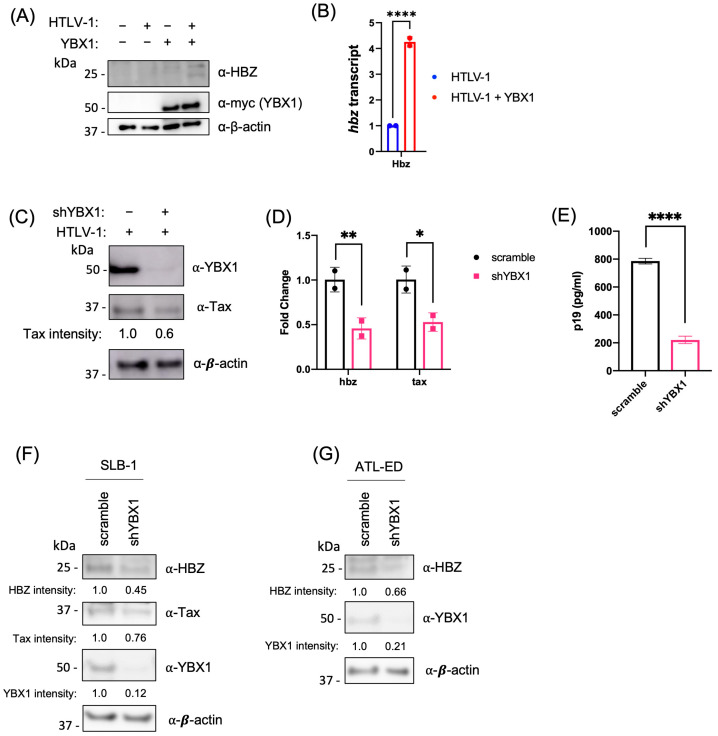
YBX1 activates sense and antisense viral transcription. (**A**) HEK293T cells were co-transfected with an HTLV-1 proviral clone and a YBX1 expression plasmid, as indicated. Protein levels were examined 48 h post-transfection by immunoblot analysis using antibodies to HBZ, myc (YBX1), and β-actin. (**B**) Quantitative RT-PCR for *hbz* and *gapdh* was performed on mRNA isolated from cells in (**A**). The total *hbz* mRNA level was determined using the ΔΔCt method, normalized to relative *gapdh* levels, and the HTLV-1 provirus alone was set at 1. (**C**) HEK293T scramble and shYBX1 cells were transfected with an HTLV-1 proviral clone for 48 h. Protein levels were examined by immunoblot analysis using antibodies to Tax, YBX1, and β-actin. Tax protein intensity relative to β-actin without YBX1 knockdown was set at 1. (**D**) Quantitative RT-PCR for *hbz*, *tax*, and *gapdh* was performed on mRNA isolated from cells in (**C**). Total *hbz* and *tax* mRNA levels were determined using the ΔΔCt method, normalized to relative *gapdh* levels, and control cells (scramble) were set at 1. (**E**) HTLV-1 gene expression was quantified from cells in (**C**) through the detection of the p19 Gag protein in the culture supernatant using ELISA. (**F**) SLB-1 and (**G**) ATL-ED transformed T-cell lines were transduced with lentiviral vectors expressing shRNA directed against YBX1. After a brief puromycin selection, cells were collected and analyzed by immunoblot using antibodies to Hbz, Tax, YBX1, and β-actin. Tax and Hbz protein intensity relative to β-actin in each scramble condition was set at 1. All graphs represent data generated from duplicate samples, and error bars represent the standard deviation (SD). The data are representative of at least three experimental repeats. Statistical significance was determined using Student’s *t*-test: * *p* ≤ 0.05, ** *p* ≤ 0.01, **** *p* ≤ 0.0001.

**Figure 5 ijms-24-13119-f005:**
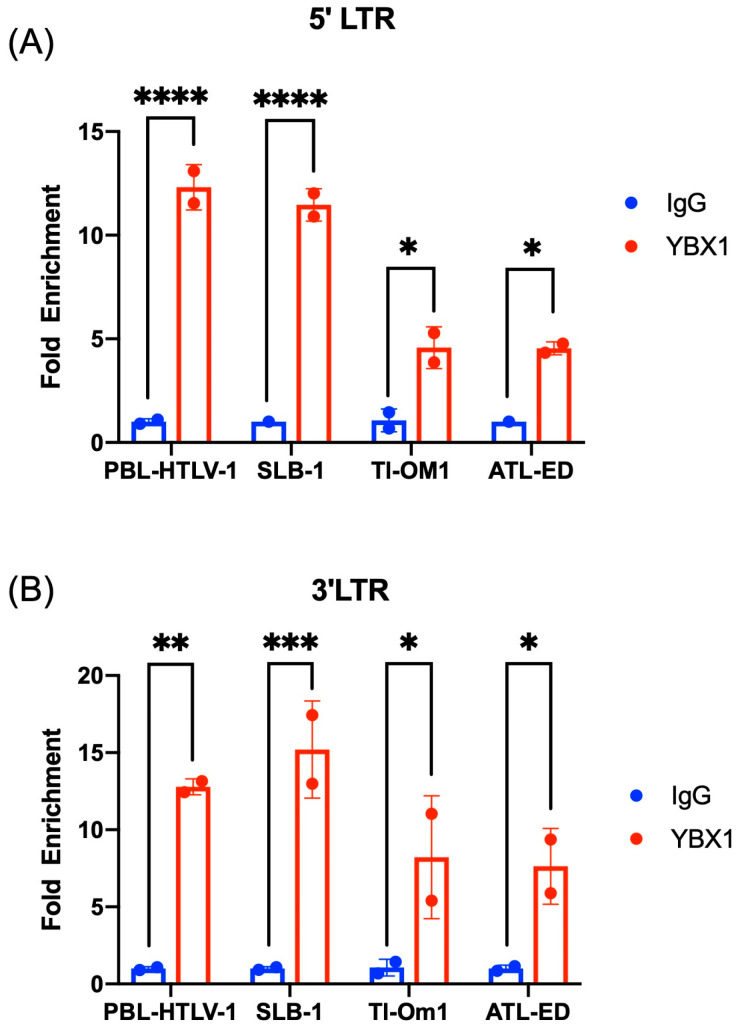
YBX1 associates with the 5′ and 3′ viral LTRs. ChIP assays were performed on cross-linked chromatin from PBL-HTLV-1, SLB-1, TL-Om1, and ATL-ED cells using either IgG or YBX1 antibodies. Retained DNA was amplified using (**A**) 5′ LTR or (**B**) 3′ LTR-specific primers and qPCR. Fold enrichment for each cell line was calculated relative to the IgG sample. Each ChIP experiment was repeated twice in duplicate. Error bars represent the standard deviation of the sample means. Statistical significance was determined using Student’s *t*-test: * *p* ≤ 0.05, ** *p* ≤ 0.01, *** *p* ≤ 0.001, **** *p* ≤ 0.0001.

**Figure 6 ijms-24-13119-f006:**
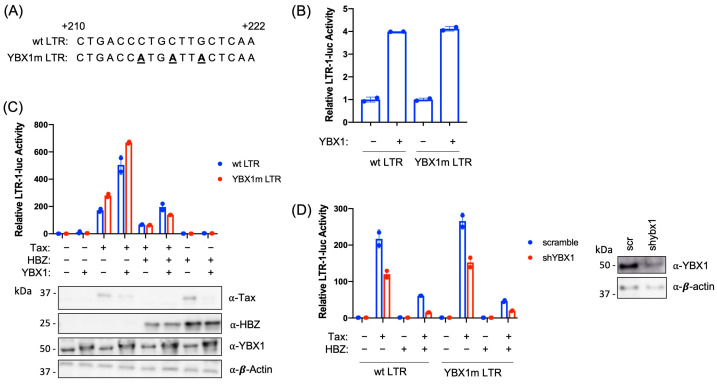
Mutation of the proposed YBX1 binding site does not affect viral transcription. (**A**) The proposed YBX1 binding site was mutated in the LTR-1-luciferase expression vector through site-directed mutagenesis (shown as bold, underlined text). An alignment of the YBX1 mutation (YBX1m) region and the wt region from the HTLV-1 LTR is shown. (**B**) HEK293T cells were co-transfected with pcDNA3.1(+) empty, YBX1, wt HTLV-1 LTR-firefly luciferase, or YBX1m HTLV-1 LTR-firefly luciferase, and tk-renilla luciferase (internal control) constructs, as indicated. At 48 h post-transfection, cells were collected for luciferase assays to measure relative LTR transactivation. The empty vector was set at 1. (**C**) HEK293T cells were co-transfected with pcDNA3.1(+) empty, YBX1, HBZ, Tax, HTLV-1 LTR-firefly luciferase, YBX1m LTR-firefly luciferase, and tk-renilla luciferase (internal control) constructs, as indicated. At 48 h post-transfection, cells were collected for luciferase assays to measure relative LTR transactivation (top panel). The empty vector was set at 1. Total cell lysates were examined by immunoblot analysis using antibodies to YBX1, HBZ, Tax, and β-actin (bottom panel). (**D**) HEK293T scramble and shYBX1 cells were co-transfected with pcDNA3.1(+) empty, Tax, HBZ, HTLV-1 LTR-firefly luciferase, YBX1m LTR-firefly luciferase, and tk-renilla luciferase (internal control) constructs, as indicated. At 48 h post-transfection, cells were collected for luciferase assays to measure relative LTR transactivation. The empty vector was set at 1. Total cell lysates were examined by immunoblot analysis using antibodies to YBX1 and β-actin (right panel). Graphs represent data generated from duplicate samples, and error bars represent the standard deviation (SD). The data are representative of at least three experimental repeats.

**Figure 7 ijms-24-13119-f007:**
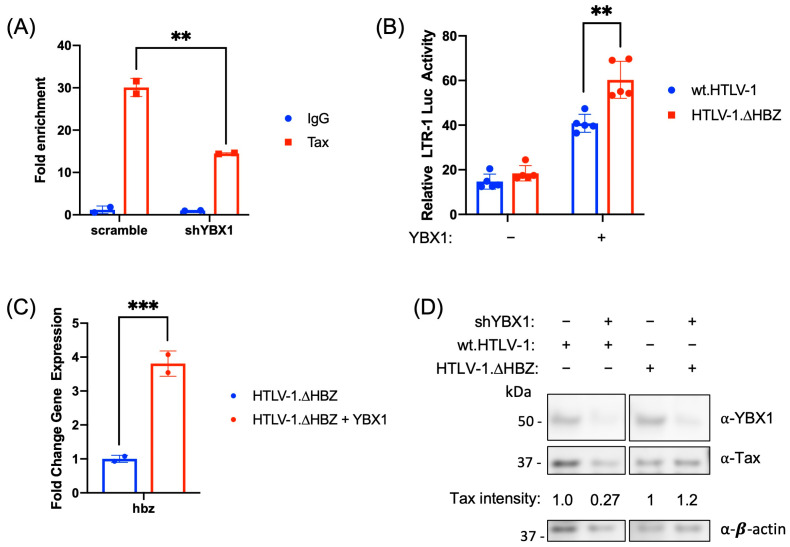
YBX1 enhances Tax binding, and HBZ represses YBX1 transcriptional activation. (**A**) SLB-1 cells were transduced with lentiviral vectors expressing shRNA directed against YBX1. After a brief puromycin selection, ChIP assays were performed on cross-linked chromatin using either IgG or Tax antibodies. Retained DNA was amplified using 5′ LTR-specific primers and qPCR. Fold enrichment was calculated relative to the IgG sample. Each ChIP experiment was repeated twice in duplicate. (**B**) HEK293T cells were co-transfected with pcDNA3.1(+) empty, YBX1, wt.HTLV-1 proviral clone, a HTLV-1 proviral clone that lacks HBZ protein (HTLV-1.∆HBZ), HTLV-1 LTR-firefly luciferase, and tk-renilla luciferase (internal control) constructs, as indicated. At 48 h post-transfection, cells were collected for luciferase assays to measure relative LTR transactivation. The empty vector was set at 1. (**C**) Total *hbz* mRNA levels from HTLV-1.∆HBZ cells in (**B**) were determined using the ΔΔCt method, normalized to relative *gapdh* levels. HTLV-1.∆HBZ provirus alone was set at 1. (**D**) HEK293T scramble and shYBX1 cells were co-transfected with wt.HTLV-1 or HTLV-1.∆HBZ proviral clones, as indicated. After 48 h, total cell lysates were examined by immunoblot analysis using antibodies to YBX1, Tax, and β-actin. Tax protein intensity relative to β-actin in each scramble condition was set at 1. Graphs represent data generated from duplicate samples (5 samples for **B**), and error bars represent the standard deviation (SD). The data are representative of at least three experimental repeats. Statistical significance was determined using Student’s *t*-test: ** *p* ≤ 0.01, *** *p* ≤ 0.001.

**Figure 8 ijms-24-13119-f008:**
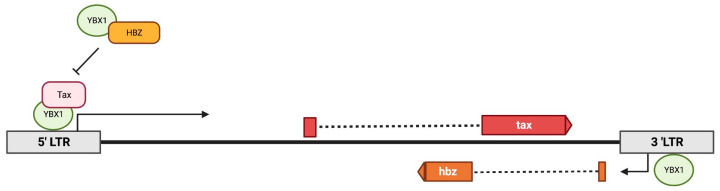
Model depicting YBX1 effects on HTLV-1 transcription. YBX1 associates with both the 5′ and 3′ LTR promoters to drive sense (*Tax*) and antisense (*Hbz*) gene expression. YBX1 can interact with and recruit Tax to the 5′ LTR, while HBZ can interact with YBX1 to block transcriptional activation.

**Table 1 ijms-24-13119-t001:** List of primer sets used for generating pCMV3 YBX1 deletion mutant constructs.

Deletion Mutant	Primer Sequence
∆A/P (∆1–55)	Forward primer: 5′-GCAACGAAGGTTTTGGGAACAGTAAAATG-3′ Reverse primer: 5′-CAAAACCTTCGTTGCGGTTCCCATGGTGG-3′
∆CSD (∆56–128)	Forward primer: 5′-GGTGGTGTTCCAGTTCAAGGCAGTAAATATG-3′ Reverse primer: 5′-GAACTGGAACACCACCGATGACCTTCTTGTC-3′
∆CTD (∆129–324)	Forward primer: 5′-CAAATGTTACAGGTCCTTAAGCGGCCGCACTC-3′ Reverse primer: 5′-GGACCTGTAACATTTGCTGCCTCCGCAC-3′

## Data Availability

Data are contained within the article or Appendix A.

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
