# Peer review of "The Pleiotropic Effects of YBX1 on HTLV-1 Transcription"

_ijms, 2023, doi:10.3390/ijms241713119_

Round 1

Reviewer 1 Report

Article entitled "The Pleiotropic Effects of YBX1 on HTLV-1 Transcription" written by Susan Smith et al. hypothesized that the HBZ/YBX1 interaction plays a key role in viral gene expression and HTLV-1 pathobiology. This paper is well written, with a detailed presentation of methods. Results and conclusions are presented satisfyingly. I do not see any disadvantages of the work and recommend the publication in its current form.

Author Response

Thank you for your time and expertise in reviewing our manuscript. We have uploaded a revised version based on other reviewer comments. 

Reviewer 2 Report

This study by Smith et al. described “The Pleiotropic Effects of YBX1 on HTLV-1 Transcription”. It is a very interesting study and my comments are below:

-In Figure 1C. I do not see any input control bands of α-HBZ?

- In Figure 2A. why there is LTR transcriptional activation inhibited only at a lower concentration, expectation is that it should decrease in a dose-dependent manner. Any explanation?

Author Response

“In Figure 1C. I do not see any input control bands of α-HBZ?”  We have updated Figure 1C with a darker exposure of the western blots for HBZ to clearly show the input lanes.

“In Figure 2A. why there is LTR transcriptional activation inhibited only at a lower concentration, expectation is that it should decrease in a dose-dependent manner. Any explanation?” In Figure 2A, we only observe a significant decrease in transcriptional activation when using lower amounts of YBX1 and a constant amount of HBZ. When we titrate the amount of YBX1 higher, we no longer observe an HBZ-mediated decrease in transcriptional activation. This is consistent with our model in Figure 8 which suggests HBZ is able to interact with YBX1 and block transcriptional activation. Increasing YBX1 levels while maintaining constant HBZ levels would mean there is simply not enough HBZ present to inhibit the higher amounts of YBX1.